# Relationships Between Cultural Factors and Motor Abilities in Autistic and Non-Autistic Children

**DOI:** 10.3390/bs15121742

**Published:** 2025-12-16

**Authors:** Lindsey Anderson, Allison Gladfelter, Milijana Buac, Waifong Catherine Cheung, Ziteng Wang, Sinan Onal

**Affiliations:** 1School of Allied Health & Communicative Disorders, Northern Illinois University, DeKalb, IL 60115, USAmbuac@niu.edu (M.B.); wcheung@niu.edu (W.C.C.); 2Department of Industrial and Systems Engineering, Northern Illinois University, DeKalb, IL 60115, USA; 3Department of Industrial Engineering, Southern Illinois University Edwardsville, Edwardsville, IL 62026, USA

**Keywords:** autism, culture, motor skills, balance, moderation, regression

## Abstract

Autism and motor abilities have been found to be closely related. Culture affects motor development as well as various diagnostic criteria for an autism diagnosis, yet cultural factors are rarely considered in research and in clinical diagnostics. This study explored the relationship between cultural factors, characteristics of autism, and balance abilities in autistic (*n* = 16) and non-autistic (*n* = 28) children by utilizing a demographic survey, the Social Responsiveness Scale, 2nd edition (SRS-2), and the Movement Assessment Battery for Children-2 (MABC-2) Balance subtest. A multiple linear regression model was used to analyze whether the relationship between autism characteristics and balance abilities still stands when cultural factors are considered, and to determine which cultural factors moderate the relationship. Moderation analyses with Holm–Bonferroni correction tested whether cultural factors altered the strength of the SRS-2 and MABC-2 balance association. The results of this study were consistent with previous research in that there is still a strong relationship between autism and balance abilities even when cultural factors are considered. The results further indicated that, in addition to autistic characteristics, age, gender/sex, and ethnic origin were statistically significant contributors to the multiple linear regression model. No significant moderation effects were detected, indicating consistency of the autism–balance relationship across cultural groups examined. In conclusion, cultural factors must be considered in autism research, as well as in the development and implementation of diagnostic and treatment protocols for autistic children.

## 1. Introduction

Autism is a neurodevelopmental difference currently seen as a disability with specific characteristics that vary for each autistic individual. Common characteristics observed in autistic children include differences in social communication, monotropic interests, sensory processing, and motor differences ([2]; [67]). For example, delayed attainment of developmental milestones, such as the onset of first words, walking, and sitting, have been previously reported in autistic children compared to neurotypical children throughout the literature ([54]; [72]). Although developmental communication and motor differences in autistic individuals are frequently reported, a recent scoping review of studies analyzing autistic walking gait discovered that not one single previous study considered cultural factors in their gait analysis ([11]). This is in spite of the fact that motor skill development is known to be influenced by one’s culture, including walking age and gait ([38], [39]). To develop representative assessments and interventions for autistic children, research on motor skills must take cultural factors into account.

### 1.1. Motor Skills as an Early Autism Indicator

Delayed language development is often the earliest concern reported by parents of children later diagnosed with autism ([72]). However, motor skill differences may appear even earlier ([22]; [42]; [43]). For example, poor postural control during a pull-to-sit task may predict autism in infants as young as six months of age ([22]). In one study, [22] ([22]) examined videos of siblings of autistic children performing the pull-to-sit task, assessing the infant’s head alignment relative to the spine. Head lag was strongly associated with later autism diagnoses, in which more children later diagnosed with autism between 30–36 months of age displayed a head lag compared to those with no developmental delays. A second study confirmed that infants more likely to be diagnosed with autism already showed a head lag at six months of age than infants with a lower likelihood of autism ([22]). Although motor differences are rarely considered when seeking a diagnosis, they may be an early indicator of autism.

Motor differences persist as autistic children age. In one study, 79% of school-aged autistic participants showed motor skill differences on the Movement Assessment Battery for Children ([29]), a standardized fine and gross motor skill assessment ([31]). Another study found 86.9% of 11,814 autistic children were at risk for developmental coordination disorder ([5]). Even when other co-occurring conditions are absent (e.g., intellectual disability, genetic or neurological disorders), widespread motor differences have still been identified in autistic children ([60]). It is worth noting that, even though motor differences are commonly found in autistic children, only 31.6% of these children reportedly receive physical therapy services ([5]), showing an urgent need for responsive motor assessment and intervention to better serve these children.

This link between autism and early motor differences contributed to the Cerebellar Sensitive Period Hypothesis of autism, which posited that cerebellar dysfunction during the neo- and peri-natal period may underlie autism ([69]). Symptoms of cerebellar disorders resemble common autistic traits, such as gait and postural differences, as well as fine and gross motor differences ([4]). Understanding cerebellar function and its role in motor control and balance in autistic children may improve assessment and intervention practices. For example, perhaps speech–language pathologists co-treating social communication skills alongside physical therapists and/or occupational therapists targeting motor skills in tandem would be more effective than therapies offered in isolation.

Despite the high prevalence of motor differences in autistic children, little is known about the influence of culture, in that most motor research in autistic individuals has not considered cultural factors (e.g., [11]). Yet, outside the autism literature, cultural factors have been shown to affect motor development in early childhood.

### 1.2. Cultural Influences on Motor Skill Development

Culture plays an important role in motor skill development, making it a necessary factor to consider when measuring motor ability. For example, sitting is one common motor developmental milestone used to determine an infant’s motor ability; however, this skill varies widely across cultures ([39]). In a study of 72 mother–infant dyads from Argentina, Cameroon, Italy, Kenya, South Korea, and the U.S., [39] ([39]) found large differences in independent sitting at five months of age. While none of the Italian infants, only 17—25% of U.S. and South Korean infants, and 25% Argentinian infants sat independently, 67% of Kenyan and 92% of Cameroonian infants demonstrated independent sitting. In just this one example, it is evident that cultural differences play a factor in when infants can be expected to achieve developmental motor milestones. Furthermore, many cultural practices, such as sleep positioning or restrictions placed on infant’s range of movement, also impact other motor milestone attainment, including crawling ([38]). It is clear that motor skill developmental norms are dependent on the individual’s culture, and it is not appropriate to use non-representative developmental norms to judge early motor skills that are not culturally responsive to the community of children they aim to assess.

### 1.3. Autism Diagnostic Criteria and Culture

The Diagnostic and Statistical Manual of Mental Disorders, Fifth Edition, Text Revision (DSM-5-TR; [2]) defines autism by differences in social-emotional reciprocity (e.g., the back-and-forth of conversations, sharing emotions), nonverbal communication (e.g., gestures, facial expressions, integrated eye contact), and in the formation or maintenance of relationships (e.g., varying behavior to a certain situation, making friendships). Although these criteria imply a sort of universal standard for social communication skills, most social behaviors are culturally determined. For example, sustained eye contact during a conversation is valued in Western cultures but may be seen as disrespectful in others, such as when children speak to adults ([16]). Similarly, cultural norms around pause time in conversation vary. Canadians and Americans may find silences lasting more than a second uncomfortable, whereas many Asian cultures accept silences that persist for a couple of minutes. In contrast, Italian and Latin American cultures often tolerate or even expect conversational overlap, including interrupting one’s conversation partner ([35], pp. 64–65). In essence, many of the social communication criteria for an autism diagnosis within the DSM-5-TR are culturally influenced.

Cultural influences are not only limited to the social communication criteria for autism in the DSM-5-TR but also extend into the restrictive or repetitive patterns of behavior criteria as well. For example, one of the DSM-5-TR criteria includes stereotyped or repetitive motor movements, such as hand flapping or rocking one’s body back and forth, usually for the autistic individual to achieve sensory regulation. However, some studies have found that autistic children in Africa did not display hand flapping or rocking, and, overall, Latino autistic adolescents and adults displayed lower levels of restrictive or repetitive behaviors compared to non-Latino autistic individuals in America ([16]), which points to cultural influences on this hallmark feature of autism. Interestingly, a review of all papers published in three autism-specific journals by [55] ([55]) found that 72% of the included studies omitted racial or ethnic descriptors of their participants. This demographic reporting problem was again confirmed by [70] ([70]). Despite clear cultural influences on core diagnostic traits, culture is rarely, if ever, considered in autism research ([11]; [55]; [70]).

Given that social communication ([16]; [28]) and motor abilities ([38], [39]) are shaped by culture, professional perceptions of autistic traits vary globally ([17]). In one worldwide survey study, 225 experts from professions that worked directly with autistic individuals, including physicians, occupational therapists, psychologists, special educators, and speech–language pathologists, answered open-ended questions pertaining to autistic characteristics as described in the International Classification of Functioning, Disability and Health ([17]). The findings showed that all professionals considered motor differences in autistic adults, but the specific aspects of differing motor skills varied. For example, only 20% of the professionals reported differences in vestibular (balance) functions and 15% reported differences in gait pattern functions, indicating that not all motor differences are perceived the same by autism professionals around the globe ([17]).

Autism researchers and professionals are not the only people whose perceptions of autism differ by culture; caregiver perceptions of autism, disability, formal diagnosis, assessment outcomes, and intervention vary cross-culturally. [16] ([16]) found that caregivers from Western cultures often seek intervention with the goal of decreasing the expressed autistic characteristics, whereas those from Eastern cultures may pursue a diagnosis in hope of a cure. Furthermore, cultural stigma impacts a family’s motivation for receiving an autism diagnosis. In some families from South Korea and India, autism’s genetic factors may negatively impact marriage prospects for the entire family, which discourages families from seeking a diagnosis ([18]; [30]). Cultural differences also affect the age at which parents first express concerns that their child may be autistic. In Western, high-income countries, parents first express concern between ages one and two, whereas in other sociocultural or socioeconomically diverse countries, concerns often were not expressed until after the second year of life ([16]). Clearly culture significantly shapes how, when, and whether caregivers pursue an autism diagnosis for their child.

### 1.4. Statement of the Problem

Although it is known that autism affects motor development ([1]; [22]; [29]; [49]; [52]; [54]) and social skills ([2]), and that culture also influences both motor development ([38], [39]) and social skill development ([16]; [28]; [35]), there are currently very limited autism research studies that consider all three factors (motor skill, social skills, and culture) together ([11]; [55]; [70]). Research involving parent perspectives from different cultures is chronically sparse ([67]), as Westernized, educated, industrialized, rich, democratic (acronymized as WEIRD) individuals are typical motor research participants ([38]). This continuous problem leads to norms that are not appropriate for many individuals on motor assessments ([47]; [59]), or on autism diagnostic assessments that are heavily dependent on social skills ([15]; [37]). Even though early motor differences, especially in balance, may be early indicators of autism and have the advantage of not requiring the child to be speaking, the absence of culturally informed research exploring motor differences in autism make their use as an early diagnostic indictor problematic. To develop and provide culturally responsive screening, assessment, and intervention tools to be used with autistic children, cultural factors and motor abilities must be considered ([11]).

### 1.5. Purpose of This Study

This study aimed to explore the underlying relationships between motor skills, characteristics of autism, and cultural factors in autistic and non-autistic children. The research questions were: (1) Does the relationship between motor abilities and autistic traits still stand when cultural differences are considered? (2) Do any cultural factors moderate the relationship between motor abilities and autistic characteristics? We hypothesized that autism characteristics and balance abilities would be closely related even when cultural factors are considered. We also hypothesized that cultural factors could play a role in the relationship between autism characteristics and balance abilities; however, given the exploratory nature of this study, we did not hypothesize which cultural factors would demonstrate significant main or moderating effects, nor the direction of these relationships.

## 2. Materials and Methods

This project utilized already collected data from a larger, ongoing study investigating the utility of walking gait differences in autistic and non-autistic children to estimate the likelihood of autism when cultural factors are considered. In the broader ongoing study, participants are video-recorded walking and complete standardized assessments of motor skills, while parents complete a lab-specific questionnaire on autism, motor skills, communication skills, and cultural factors, as well as a standardized questionnaire on social responsiveness. The present study analyzed a subset of the data from this broader study, focusing on standardized assessments of balance, social responsiveness, and cultural factors.

### 2.1. Participants

To statistically estimate how many participants were necessary to include in this exploratory study, a power analysis was conducted using G*Power software, version 3.1.9.7 ([12]). Using an alpha level of 0.05, power of 0.80, and a moderate effect size of 0.25 as the set parameters for the planned multiple linear regression model with one dependent variable (i.e., the Movement Assessment Battery for Children-2 (MABC-2) balance subscale standard scores) and eight total predictors (i.e., the Social Responsiveness Scale-2 (SRS-2) overall standard score, SRS-2 Social Communication Index, SRS-2 Restrictive and Repetitive Behaviors Score, age, gender/sex, socioeconomic (SES) level, number of known languages, and race/ethnicity), the minimum total sample size needed was 34. As such, the currently available sample size of 44 participants from the broader ongoing study was considered appropriate for this exploratory study.

To be included in the broader research study, all participants were between the ages of 5–12, had the ability to walk independently, and were not taking any medications that would impact movement ([19]). To better understand the relationship between motor skills and autism specifically, participants representing a wide range of neurotypes were recruited for this study. This is especially important because nearly 91% of autistic children report co-occurring psychological conditions ([50]), a rate much higher than the general population ([48]). As such, autistic and non-autistic participants with co-occurring conditions (e.g., Attention-deficit/hyperactivity disorder (ADHD), anxiety disorders, obsessive compulsive disorder (OCD)) were included. Of the 44 participants included in this study, 16 children had a formal diagnosis of autism, with 9 reporting co-occurring conditions of ADHD, OCD, anxiety disorder, asthma, dysgraphia, fine motor developmental delay, sensory processing disorder, SATB2-associated syndrome, and precocious puberty. Of the remaining 28 non-autistic children, 6 reported conditions of ADHD, sensory processing disorder, anxiety disorder, auditory processing disorder, non-verbal learning disability, ocular motor aphasia, and dyslexia. Finally, 22 of the non-autistic participants reported typical development with no known co-occurring conditions.

The Autism Diagnostic Observation Schedule, 2nd edition (ADOS-2) was not used for inclusionary purposes because it is known to have race and gender/sex biases ([15]; [37]), and because it is possible for children with other diagnoses, such as developmental language disorder, to also score within the autistic range ([44]). Historically, autistic female participants have been excluded from autism research when using the ADOS as a confirmatory measure for diagnosis, so instead, parent reported diagnosis was the basis for recruitment of autistic participants ([15]). To get an estimate of each individual’s autistic characteristics, the Social Responsiveness Scale, 2nd edition (SRS-2) was administered to all participants (autistic and non-autistic; [14]). The SRS-2 is a reliable questionnaire that includes separate male and female forms, and its estimates of autism characteristics align with autism diagnostic criteria ([24]). Participants came from a variety of cultural backgrounds, such as gender/sex, ethnic origin, cultural identity, and languages spoken in the home. All participants came from a Midwestern region of the United States.

### 2.2. Materials

The three main factors of interest in this study were (1) autism-related characteristics, (2) motor skills, primarily in the area of balance, and (3) cultural factors. The materials used to measure these three factors are discussed in more detail below.

#### 2.2.1. Social Responsiveness Scale, Second Edition (SRS-2)

Caregivers completed the SRS-2 to determine the extent to which their child displayed autistic characteristics. The autism estimations made by the SRS-2 have been shown to be compatible with DSM criteria for an autism diagnosis ([24]). Statistically, one benefit of the SRS-2 is that it assesses a child’s autistic characteristics relative to a population sample using a *T*-score, a continuous variable, rather than a binary cutoff score (i.e., autistic vs. non-autistic), a categorical variable. The use of this continuous variable then allowed for a direct comparison of the extent of autistic characteristics across all 44 participants included in this study, regardless of diagnostic status.

Because the SRS-2 shows high diagnostic sensitivity (92%) and specificity (92%) and can be administered to the caregiver in approximately 15 min ([10]; [14]), it was selected to estimate participants’ autistic characteristics rather than administering a more time intensive measure (e.g., ADOS-2) to prevent the child from potentially experiencing fatiguing during the motor assessment. Also, historically, females have been excluded from participation in research when the ADOS-2 was used as inclusionary criteria as it was predominantly normed on males and has shown some gender/sex bias ([15]; [37]). The SRS-2 was normed on American individuals. The normative sample aligned with the U.S. Census of 2010 in terms of proportions race/ethnicity, geographic location, and parent education level, making it a representative sample as of 2010. The SRS-2 includes questions about how the child communicates and interacts with others (i.e., topic maintenance, walking in between two people talking), as well as how they interact with the world around them (i.e., range of interests, sensory sensitivities). Female and Male forms for the SRS-2 were available for use, as well as forms in both English and Spanish. The SRS-2 protocols were all scored by trained graduate research assistants in the fields of speech–language pathology or physical therapy.

#### 2.2.2. Movement Assessment Battery for Children, Second Edition (MABC-2)

The Balance subscale items of the MABC-2 ([31]) were administered to the children participating in this study. Autistic children have previously been shown to significantly differ from their neurotypical peers on the MABC-2 overall ([29]) and on static balance subcomponent skill in particular ([71]). The balance subtest items have been shown to be highly related to characteristics of autism ([1]; [27]; [46]). The first balance task has the child stand on one leg on a mat or board depending on age with their arms at their side. The children are asked to stand for at least 30 s on each leg in this manner. The timer is stopped when there is a fault. The second balance task requires the child to walk along a line with their heels raised or with the heel of one foot against the toes of the other foot, depending on their age. The final balance task requires the child to make five consecutive jumps from mat-to-mat landing in a balanced position on one or two feet depending on their age. All MABC-2 balance tasks were administered and scored by trained graduate assistants in the fields of speech–language pathology or physical therapy.

#### 2.2.3. Cultural Factor Survey

Caregivers completed a background questionnaire about their child through Qualtrics, an online software used to collect and analyze surveys. According to the American Psychological Association (APA) cultural influences include variables such as “age, generation, gender/sex, ethnicity, race, religion, spirituality, language, sexual orientation, gender identity, social class, ability/disability status, national origin, immigration status, and historical as well as ongoing experiences of marginalization, among other variables” ([3], p. 8). The questionnaire included questions about the child’s culture, medical history, family history, speech and motor developmental milestones, daily behavior of the child, and caregiver concerns. Questions asked about gender/sex, ethnic origin, cultural identity, caregiver years of education, and languages spoken. Language history questions included milestones of babbling, first words, sentences, as well as comprehension of simple directions. Medical history questions included birth history, current health status, and handedness. Family history included listing any relatives with speech, language, reading, or hearing difficulties. Daily behavior questions explored how the child interacts with others, favorite toys/activities, and strengths. Caregiver concerns regarding speech, language, and hearing differences or difficulties in their child were also collected, as well as information on whether the child received any form of developmental therapy and what the outcomes of the therapy were. Caregivers were asked to rate their child’s ability to speak, understand, and read English and any other home languages to further identify language proficiency. These surveys were provided in both English and Spanish.

### 2.3. Procedure

All recruitment and experimental procedures were implemented in accordance with the approved university Internal Review Board (IRB) protocols. Recruitment of participants for the broader gait study was done largely through the distribution of flyers throughout the community. Support letters from autism day schools, community programs, and clinics local to Northern Illinois University (NIU) were obtained to aid in the distribution of such flyers. Information about the study was then distributed via schoolwide e-mails sent to families, flyers distributed to children at their schools, distributed at community events, posted in public spaces such as libraries, distributed to local clinicians, and through databases available to the research team.

Once participants agreed to join the study, caregivers were provided with a consent form. A research assistant explained each section of the consent form and checked for understanding and asked if the family had any questions. If the caregiver agreed to participate, they signed the consent form. Then, the child participants were asked to provide assent prior to participating in the study. For all child participants, a social story format was used for collecting the child’s assent to participate, which is a supportive way to share information about upcoming events with autistic children to ease transitions and enhance understanding. Both caregivers and children were informed that participation in the study was voluntary, and they could withdraw at any time.

After consent and assent were given, the demographic and cultural factor survey was completed by the caregiver on a password-protected tablet or laptop computer that was provided to them within the research lab. The expected completion time for the survey was approximately 10 min. Then, the SRS-2 was administered to the caregiver by a trained graduate assistant. The SRS-2 is a paper and pencil questionnaire with an expected completion time of approximately 15 min.

While the caregiver completed the survey and SRS-2, the child completed the Balance subtest items of the MABC-2. The completion time of the Balance subtest was approximately 10 min. The graduate assistant administering the MABC-2 recorded the results on the test form. In total, the combination of all three assessments collected for this study took approximately 25 min to complete. Upon completion of the session, the children were given a small toy as compensation for their time.

### 2.4. Descriptive Statistics

To gather an overall picture of the participants in this study, descriptive statistics were calculated to identify the mean, standard deviation (SD), minimum, and maximum values of the participants’ age, socioeconomic status (SES; as estimated by the caregivers’ total years of education), MABC-2 Balance standard score, SRS-2 *T*-scores, SRS-2 SCI scores, and SRS-2 RRB scores using SPSS version 26.0 ([36]). Descriptive statistics regarding the number of participants of each gender/sex, ethnic origin category, cultural identity, number of languages used, and which languages used were also calculated. The key descriptive statistics for all participants together are summarized in Table 1. Mann–Whitney U tests confirmed that groups did not differ significantly in age (*p* = 0.171) or SES (*p* = 0.213) but differed as expected in SRS-2 scores (*p* < 0.001) and MABC-2 Balance performance (*p* = 0.025).

#### 2.4.1. Age

The mean age of participants was 8.23 years (SD = 2.25 years, ranging from 5 to 12). Of the 16 autistic participants, the mean age was 8.81 years (SD = 2.17) and ranged from 6 to 12 years. Of the 28 non-autistic participants, the mean age was 7.89 years (SD = 2.27) and ranged from 5 to 12 years.

#### 2.4.2. Gender/Sex

Of the 44 total participants, 27 participants identified as male, 15 as female, one as nonbinary, and one did not report preferred gender pronouns. Of the 16 autistic participants, more identified as males than any other gender/sex (13 males, one female, one nonbinary, one did not report). Of the 28 non-autistic participants, 14 identified as males and 14 identified as females.

#### 2.4.3. Socioeconomic Status (SES)

The average years of education of Caregiver 1 was 17.45 (SD = 6.41) with a range of 0 to 33 years and Caregiver 2 was 13.41 (SD = 5.88) with a range of 0 to 23. Of the 16 autistic participants, the average years of education of Caregiver 1 was 15.75 years (SD = 3.57). Caregiver 2 was 14.06 (SD = 4.40). Of the 28 non-autistic participants, the average years of education of Caregiver 1 was 18.43 years (SD = 7.47) and Caregiver 2 was 13.04 (SD = 6.63). In other words, the caregivers, on average, had some education past high school across the autistic and the non-autistic children.

#### 2.4.4. SRS-2 *T*-Scores

SRS-2 Total *T*-scores of 59 or lower are considered “within normal limits” according to the administration manual. Across all children (both autistic and non-autistic combined), the average SRS-2 overall *T*-score was 61 (SD = 14, ranging from 39 to 90). Of the 16 participants previously diagnosed with autism, the average SRS-2 overall *T*-score was 72 (SD = 11, ranging from 52 to 90), which is displaying significant autistic characteristics. Of the 28 non-autistic participants, the average SRS-2 overall *T*-score was 54 (SD = 11, ranging from 39 to 79), which is not displaying significant autistic characteristics. It is worth noting that some of the autistic children scored below the autistic range of scores on the SRS-2, and that some non-autistic children scored within the autistic ranges of scores.

The two DSM-5 Compatible Scales on the SRS-2, namely the Social Communication and Interaction (SCI) subscale and the Restrictive and Repetitive Behaviors (RRB) subscales, were also analyzed. The SRS-2 SCI average *T*-score across all children (both autistic and non-autistic combined), was 59.80 (SD = 13.45, ranging from 39 to 90). Of the 16 participants previously diagnosed with autism, the average SRS-2 SCI *T*-score was 70.69 (SD = 10.80, ranging from 52 to 90). Of the 28 non-autistic participants, the average SRS-2 SCI *T*-score was 52.74 (SD = 10.62, ranging from 39 to 76). The average SRS-2 RRB *T*-score was 64.11 (SD = 16.03, ranging from 41 to 90). Of the 16 participants previously diagnosed with autism, the average SRS-2 RRB *T*-score was 77.19 (SD = 10.04, ranging from 54 to 90). Of the 28 non-autistic participants, the average SRS-2 RRB *T*-score was 56.64 (SD = 13.93, ranging from 41 to 88).

#### 2.4.5. MABC-2 Balance Scores

The MABC-2 uses standardized scaled scores to compare performance to the normative data. A standard scaled score of 10 is the mean (SD = +/−3), with scores from 7–13 considered within the typical range, scores 6 and below are considered below the typical range, and scores above 13 are considered above the typical range. The average MABC-2 Balance standard score was 6.50 (SD = 4.15, ranging from 1 to 16), which is considered below the typical range. Of the 16 autistic participants, the average MABC-2 Balance standard score was 4.65 (SD = 3.36, ranging from 1–11), which is considered below the typical range. Of the 28 non-autistic children, the average MABC-2 Balance standard score was 7.57 (SD = 4.23, ranging from 1 to 16), which is considered within the typical range.

#### 2.4.6. Ethnic Origin

Of the participants who only selected one ethnic origin, 28 selected white, six selected Hispanic, and three selected Black. Of the participants who selected multiple ethnic origins, three selected Hispanic and white, three selected Asian or Pacific Islander and white, and one selected American Indian or Alaskan Native and white.

#### 2.4.7. Cultural Identity

The following information was not included in the final regression model because of conceptual similarity to ethnic origin and number of languages used but is provided to give a more detailed background of the study participants. Nine participants did not provide a cultural identity. Of the participants who reported one cultural identity, 20 identified as U.S. American, one identified as Midwestern American, one identified as Mexican, one identified as Catholic, and one identified as Christian. Of the participants who provided more than one cultural identity, four identified as U.S. American and Mexican, one identified as U.S. American and Greek and Italian, one identified as U.S. American and Caribbean and African American, one identified as U.S. American and Roman Catholic, two identified as U.S. American and German, one identified as Mexican/Hispanic and white, and two identified as U.S. American and Polish and Catholic.

#### 2.4.8. Number of Languages

Thirty-nine participants reported that they used one language, and five participants used two languages. Of the 16 autistic participants, 15 used one language, and one used two languages. Of the 28 non-autistic participants, 24 used one language, and four used two languages. For participants who did not respond to the number of languages used/exposed to in the home, the value of 1 language was the default. It was assumed that everyone uses or has been exposed to at least 1 language.

#### 2.4.9. Languages Used

This information is provided to give a more detailed picture of the participants in the study but was not included in the final regression model because of the conceptual similarity and the high statistically significant correlation with the number of languages used. Of the 16 autistic participants, 15 used English and 1 did not respond. Of the 28 non-autistic participants, 22 used English, 2 used German and English, 2 used Spanish, and 2 did not respond.

### 2.5. Statistical Analyses

In this project, we aimed to address two research questions: Does the relationship between motor abilities and autistic characteristics still stand when cultural differences are considered? Which, if any, cultural factors moderate the relationship between motor abilities and autistic characteristics the most? To answer these questions, we conducted a regression analysis, in which regression coefficients with the confidence intervals set to 95%, estimates of the model fit (R and R^2^), and statistical significance of the independent variables (*t*-values and *p*-values) were all extracted from the analysis. Independent variable coefficients with a *p* value of less than 0.05 were considered statistically significant. Analyses were conducted in Python 3.12 (Google Colab).

#### 2.5.1. Establishing the Final Set of Independent Variables

First, a Pearson’s bivariate correlation analysis was used to identify any potential collinearity of independent variables for the later planned regression analysis. When any independent variables conceptually captured similar information and were significantly correlated with one another (i.e., *p* < 0.05), one representative variable was selected for inclusion in the planned regression analysis. Using this approach, the SRS-2 *T*-score (which is a combination of the SCI and RRB subscale scores) had a strong, positive relationship with the SRS-2 SCI score (*r* = 0.989, *p* < 0.001) and the SRS-2 RRB score (*r* = 0.912, *p* < 0.001). Because all three SRS-2 scores conceptually reflected a participant’s autistic traits and were highly correlated, only the SRS-2 Overall *T*-score was included in the regression analysis. Similarly, C1 years of education had a moderate, positive relationship with C2 years of education (*r* = 0.541, *p* < 0.001), and both were conceptually intended to estimate participants’ SES level. Therefore, only the C1 years of education was used in the regression model. Finally, cultural identity had a moderate, positive relationship with the number of languages spoken in the home (*r* = 0.599, *p* < 0.001). This finding, although not anticipated, was perhaps unsurprising, as many of the participants who reported using two or more languages at home also reported two or more cultural identities that were conceptually related to their languages. For example, two participants identified as U.S. American and German and reported using two languages (English and German) at home. As such, only the number of languages was used in the regression model.

After eliminating the conceptually similar and significantly correlated variables, the final set of six independent variables for the planned multiple regression analysis to predict the relationship between MABC-2 balance scores (dependent variable), included: the SRS-2 Overall *T*-scores (IV 1), age (IV2), gender/sex (IV3), ethnic origin (IV4), SES level based on Caregiver 1’s years of education (IV5), and number of languages (IV6).

#### 2.5.2. Variable Preparation and Coding

To keep predictors on a common scale and make coefficients easier to read, the continuous variables, including age (years), SRS-2 Overall *T*-score, Caregiver 1 education (years), and number of languages spoken, were standardized to *z*-scores (mean 0, SD 1) before modeling. This standardization puts predictors measured on different scales onto the same metric, making effects comparable; in the regression, each coefficient for a *z*-scored predictor represents the expected change in MABC-2 Balance score associated with a 1–SD increase in that predictor (e.g., if *β* = −3.00 for SRS-2_*z*, a one–SD higher SRS-2 is associated with a 3-point lower Balance score, holding other variables constant).

Because several gender/sex categories had very small counts, gender/sex was grouped as male (reference) vs. non-male (female, non-binary, or not reported) and entered as a dummy variable (Gender_nonMale = 1 for non-male, 0 for male). Using this coding, the data comprise 27 male and 17 non-male participants; the Gender_nonMale coefficient is interpreted as the adjusted mean difference in MABC-2 Balance scores for non-male vs. male (reference). That is, a positive value indicates higher MABC-2 Balance scores among non-male participants, holding all other predictors constant.

Because participants could report more than one heritage, ethnic origin was represented with three yes/no dummy variables: white, Hispanic, and Other heritage. “Other heritage” pools smaller groups in the sample (e.g., Black, Asian/Pacific Islander, American Indian/Alaska Native, and multi-heritage except Hispanic/white). These heritage indicators are not mutually exclusive (e.g., a child can be “yes” for both Hispanic and white). All three were included in the model at the same time and no reference group was selected. Across the sample (*N* = 44), 35 children endorsed white heritage, 10 endorsed Hispanic heritage, and 7 endorsed another (pooled) heritage; these counts are not mutually exclusive. In the regression, each heritage coefficient represents the adjusted mean difference in MABC-2 Balance scores for children with that heritage versus those without it (controlling for the other predictors and heritage dummies); for multi-heritage children (e.g., Hispanic and white), the implied effect is the sum of the relevant coefficients.

#### 2.5.3. Regression Model

We fit an ordinary least squares (OLS) multiple linear regression with MABC-2 Balance score as the dependent variable and eight predictor: Age (*z*), SRS-2 Overall T (*z*), Caregiver 1 education (*z*), number of languages (*z*), Gender_nonMale (dummy: 1 = non-male; 0 = male), and Ethnic origin_heritage dummy variables (white, Hispanic, Other heritage; yes/no, not mutually exclusive).

For each predictor, we report coefficients (*β*; unstandardized for dummy variables and standardized for *z*-scored continuous variables) with 95% confidence intervals and *t*-tests with *p*-values. We also report model R^2^/adjusted R^2^. For *z*-scored predictors (age, SRS-2 score, Caregiver 1 years of education, number of languages, gender/sex, and ethnic origin), *β* is the expected change in MABC-2 Balance for a 1-SD increase in that predictor (holding others constant). For dummy variable Gender_nonMale, *β* is the adjusted mean difference between the nonMale group and the reference Male group. For Ethnic origin_heritage dummy variables, coefficients reflect the association of having that heritage. Because heritage indicators can co-occur (e.g., Hispanic and white), they are not contrasts among mutually exclusive categories.

#### 2.5.4. Correlation and Multicollinearity Analysis

Because several cultural indicators were conceptually and empirically interrelated, we conducted correlation and multicollinearity diagnostics to ensure model validity. Intercorrelations among all predictors were examined (reported in Section A.1), with particular attention to cultural variables. Variance inflation factors (VIF) were calculated for all predictors in the baseline model. VIF values below 5 indicate acceptable multicollinearity, values between 5 to 10 suggest moderate concern, and values above 10 indicate severe multicollinearity requiring remediation ([9]; [51]). All VIFs in our baseline model were below 1.6, confirming that multicollinearity did not materially distort regression estimates (see Section A.2).

#### 2.5.5. Independence of Observations

For the multiple regression model, the independence of observations was checked using the Durbin–Watson statistic for autocorrelation. Only values between 1.5–2.5 were considered sufficiently independent. The Durbin–Watson statistic calculated for this model was 2.26, indicating the variables were sufficiently independent to run the multiple regression model.

#### 2.5.6. Check for Linearity

Furthermore, to check that a linear relationship between the dependent variable (MABC-2 balance subscale scores) and the primary predictive variable (SRS-2 Overall *T*-score) was present, a scatterplot in Microsoft Excel was first created and visually inspected for linearity (Figure 1). The scatterplot visually revealed a linear relationship.

#### 2.5.7. Moderation Analyses

To test whether cultural factors moderate the association between autistic characteristics and balance, we conducted interaction analyses following recommendations for multiple testing correction ([33]). Specifically, we tested six pre-specified interactions between SRS-2 Overall *T*-score (standardized) and each cultural moderator: Gender/sex (nonMale vs. Male; dummy-coded); ethnic origin heritage indicators (white, Hispanic, Other; multi-label binary coding as described above); socioeconomic status (Caregiver 1 years of education; standardized); and number of languages spoken in the home (standardized).

Each interaction was tested in a separate regression model that included all baseline predictors (SRS-2, Age, Gender/sex, ethnic origin heritage indicators, SES, and number of languages) plus one SRS-2 × Moderator interaction term. For each test, we report the unstandardized coefficient (*β*), 95% confidence interval, *p*-value, change in explained variance (Δ*R*^2^) relative to the baseline model, and the model *R*^2^. To control family-wise error rate across these six pre-specified tests, we applied Holm–Bonferroni sequential correction with α = 0.05 ([33]). This method adjusts *p*-values to account for multiple comparisons while maintaining greater statistical power than simple Bonferroni correction.

Additionally, we fit an exploratory omnibus model including all six interactions simultaneously (reported in Section A.3). This model is clearly labeled as exploratory given power constraints with *N* = 44 and 14 predictors. We calculated variance inflation factors (VIF) for this model to assess multicollinearity among interaction terms.

A sensitivity analysis was conducted to determine detectable effect sizes. With *N* = 44, eight baseline predictors and one interaction term, α = 0.05, and power = 0.80, we were adequately powered to detect moderate-to-large interaction effects (*f*^2^ ≥ 0.24) but likely underpowered for small-to-moderate effects (*f*^2^ < 0.15). Accordingly, moderation findings should be interpreted as preliminary evidence requiring replication in adequately powered samples.

### 2.6. Use of Artificial Intelligence

In this study, we used Claude AI’s Sonnet 4.5 model for the purposes of code generation, data visualization, and result interpretation. We prompted the AI to enhance the Python code we created for data processing and statistical analysis, generating a variety of numerical, textual, and graphical outputs for better visualization of the results. We also used AI to assist in interpreting the statistical analysis results based on our initial observations and methodology. We have reviewed and edited the AI’s output and take full responsibility for the content of this publication.

## 3. Results

### 3.1. Multiple Linear Regression Model (Research Question 1)

The multiple regression analysis assisted in determining the contribution of independent variables (i.e., autistic traits and cultural measures) towards explaining the variability in the dependent variable, in this case, MABC-2 Balance scores. The independent variables together explained approximately 67% (*F*(8, 35) = 8.899, *p* < 0.001, *R*^2^ = 0.67, adjusted *R*^2^ = 0.60) of the variability in the MABC-2 Balance scores.

The unstandardized coefficients (*β*_1_) generated from the model indicate how an independent variable is associated with the dependent variable to change while the other independent variables are held constant. The statistical significance of each of the independent variables indicates whether or not each of the independent variables significantly added to the prediction. In this model, five variables added statistically significantly to the prediction: SRS-2 Overall *T*-score (*β* = −3.00; *p* < 0.001), age (*β* = 1.61; *p* < 0.01), gender_nonMale (*β* = 2.92; *p* < 0.01), ethnic origin_white heritage (*β* = 5.61; *p* < 0.001), and ethnic origin_Hispanic heritage (*β* = 4.87; *p* < 0.001). A summary of the regression coefficients for all of the independent variables is summarized in Table 2.

#### 3.1.1. SRS-2 Overall *T*-Scores

The SRS-2 Overall *T*-score was a significant predictor in this model (standardized *β* = −3.00; *p* < 0.001), in which for each one standard deviation (14.05) increase in the *T*-score, there is a decrease in the MABC-2 Balance score by 3.00 points, which is statistically significant when all other independent variables are held constant.

#### 3.1.2. Age

Age was also a significant predictor in this model (*β* = 1.61; *p* < 0.01), in that for each one standard deviation (2.25) increase in age, there is an increase in the MABC-2 Balance score by 1.61 points, which is statistically significant when all other independent variables are held constant.

#### 3.1.3. Gender/Sex

The effect of gender_nonMale in this model (*β* = 2.92; *p* < 0.01) indicated that for non-male individuals, compared to male, there is an increase in the MABC-2 Balance score by 2.92 points, which is statistically significant when all other independent variables are held constant.

#### 3.1.4. Ethnic Origin

The effect of ethnic origin_white heritage in this model (*β* = 5.61; *p* < 0.001) means that for children with white heritage, compared to those without, there is an increase in the MABC-2 Balance score by 5.61 points, which is statistically significant when all other independent variables are held constant. The effect of ethnic origin_Hispanic heritage in this model (*β* = 4.87; *p* < 0.001) means that for children with Hispanic heritage, compared to those without, there is an increase in the MABC-2 Balance score by 4.87 points, which is statistically significant when all other independent variables are held constant. For multi-heritage children (e.g., those reporting both Hispanic and white heritage, *n* = 3), the predicted balance advantage reflects the combined effects of both heritage coefficients (Hispanic: *β* = 4.87 + white: *β* = 5.61 = 10.48 points), though this estimate is imprecise given small cell sizes.

#### 3.1.5. Other Measures

Other variables hypothesized to be contributors to the model were not considered statistically significant in this study, including Ethnic origin_Other heritage (*β* = 2.05; *p* = 0.12), SES as measured by C1 years of education (*β* = 0.04; *p* = 0.94), and number of languages (*β* = −0.21; *p* = 0.62).

### 3.2. Moderation by Cultural Factors (Research Question 2)

To address whether cultural factors moderate the relationship between autistic characteristics and balance abilities, we tested six pre-specified interactions between SRS-2 Overall *T*-score and cultural moderators (Table 3). After Holm–Bonferroni correction for multiple comparisons (family-wise α = 0.05), none of the tested interactions reached statistical significance (all *p*_adjusted = 1.000; Table 3). These null interactions suggest that, although cultural factors influence balance performance overall (see Section 3.1), they do not meaningfully alter how autistic characteristics predict balance within this sample.

Specifically, the SRS-2 × Gender/sex interaction (*β* = −1.11, 95% CI [−3.35, 1.14], *p*_adjusted = 1.000, Δ*R*^2^ = 0.009) did not significantly predict balance scores. This suggests that the negative relationship between SRS-2 scores and balance performance was similar for males and non-males. Similarly, interactions with ethnic origin heritage indicators were non-significant: white heritage (*β* = 0.16, 95% CI [−2.16, 2.47], *p*_adjusted = 1.000, Δ*R*^2^ < 0.001), Hispanic heritage (*β* = −0.70, 95% CI [−2.89, 1.49], *p*_adjusted = 1.000, Δ*R*^2^ = 0.004), and Other heritage (*β* = −0.81, 95% CI [−3.78, 2.16], *p*_adjusted = 1.000, Δ*R*^2^ = 0.003). Interactions with socioeconomic status (*β* = −0.46, 95% CI [−1.50, 0.58], *p*_adjusted = 1.000, Δ*R*^2^ = 0.008) and number of languages (*β* = −0.55, 95% CI [−1.46, 0.35], *p*_adjusted = 1.000, Δ*R*^2^ = 0.014) were also non-significant.

These results indicated that, within the statistical power available (adequate for moderate-to-large effects; see Section 2) and the range of cultural variation in this Midwestern U.S. sample, the negative association between SRS-2 scores and balance performance (reported in Section 3.1) was largely consistent across the cultural factors examined. The small Δ*R*^2^ values (range: 0.000–0.014) suggested that interaction terms contributed minimally to model fit beyond main effects.

An exploratory omnibus model including all six interactions simultaneously (Section A.3) also showed no significant interactions (all *p* > 0.07), corroborating the one-by-one tests. However, this omnibus model had limited degrees of freedom (*N* = 44, 14 predictors) and should be interpreted cautiously. VIF values in the omnibus model ranged from 1.16 to 3.59, with most below 2.0, indicating acceptable multicollinearity even in the saturated model (Section A.4).

## 4. Discussion

In the current study, we predicted that autism characteristics and balance abilities would be closely related. We also predicted that cultural factors, such as gender/sex, ethnic origin, SES, and number of languages, would play a role in the relationship between autism characteristics and balance abilities. The results indicated a strong relationship between autism characteristics, measured by SRS-2 *T*-scores, and balance abilities, measured by MABC-2 Balance subscale scores. SRS-2 overall *T*-scores, age, gender/sex, and ethnic origin were found to significantly affect MABC-2 Balance scores. While this study was consistent with previous research finding a significant relationship between balance abilities and autism, cultural factors, such as age, gender/sex, and ethnic origin were also shown to play a role in balance outcomes. The results are further discussed below.

### 4.1. Main Effects of Autism Characteristics and Cultural Factors on Balance Abilities

#### 4.1.1. Autism Characteristics and Balance Abilities

One of the aims of this study was to determine whether the relationship between balance abilities and autism characteristics still stood when cultural factors were considered. Consistent with previous research ([1]; [6]; [22]; [27]; [29]; [60]), we found a strong, negative relationship between SRS-2 and MABC-2 Balance scores, meaning that as participants presented with more autistic characteristics, they exhibited more difficulty on balance-based motor tasks.

The Cerebellar Sensitive Period Hypothesis of Autism offers a possible explanation for this relationship ([69]). According to the Cerebellar Sensitive Period Hypothesis, the relationship between autism characteristics and motor abilities both stem from neo- and peri-natal cerebellar dysfunction and differences in neural circuitry that manifest over time, via developmental diaschisis, into autism ([69]). There is evidence of some potential cellular differences in the cerebellum of autistic children that may affect social behaviors, including fewer Purkinje cells and less gray matter ([65]). Additionally, patients with cerebellar injury sometimes present with social and linguistic behaviors similar to those found in autistic individuals ([69]). These cerebellar differences may help explain the social differences that serve as autism diagnostic criteria of the DMS-5-TR ([2]).

The cerebellum is also considered a key neurological area for balance ([41]; [45]). Balance impairment is often a recognizable symptom in patients with lesions in specific areas of the cerebellum ([41]). Ataxia, a neurological condition that affects balance, arises from cerebellar problems, and many ataxic characteristics can be categorized as having balance difficulties ([45]). For example, ataxic gait is characterized as having a widened base as a compensatory strategy for balance difficulties while walking ([45]). If cerebellar development is disrupted early in life, as posited in the Cerebellar Sensitive Period hypothesis, the autistic child’s social skills and motor skills would both be affected, which is consistent with the findings in this study, as well as in previous research (e.g., [6]; [27]; [60]).

Unlike social differences, motor difficulties are not part of the autism diagnostic criteria, per the DSM-5-TR, and physical therapists are not often included in diagnostic teams ([26]). A multi-disciplinary team approach to autism diagnostics is considered best practice, including a variety of professionals to assess the various diagnostic areas outlined in the DSM-5-TR. In a survey for speech–language pathologists regarding co-occurring conditions with autism and members of the diagnostic team, motor impairment was not an included option and physical therapists were not listed as team members to consider ([26]), both of which reflect the limited understanding of how social skills and motor skills intertwine in the broader literature. With increasing research reporting motor differences in a large portion of autistic individuals, physical therapists should be considered as additional professionals on autism diagnostic teams to help support diagnostic decisions as well as make recommendations for future treatment areas.

As established by [5] ([5]), only about a third of autistic children who would benefit from physical therapy services receive them. Consistent with the low scores on the MABC-2 balance subscales in our sample, referrals for physical therapy assessments should be routinely considered for autistic children, as well as collaborations with physical therapists during speech therapy sessions. According to one scoping review, 12 out of 13 interventions that targeted motor skills also improved language outcomes in autistic children–pointing to the potential benefits of speech–language pathologists and physical therapists cotreating rather than providing services in isolation ([53]). Therapy sessions that focus on the “whole child,” by targeting more than just one domain generally reveal marked improvements in the child’s abilities.

Complex motor skills are used in play with peers, and as such, help to develop friendships ([7]). If a speech–language pathologist is working on social language skills with an autistic child, but not collaborating with a physical therapist on the child’s motor skills, the child may not be able to successfully practice newly acquired social skills on the playground or in the classroom if weak motor skills are limiting their participation in these activities with other children. If the speech–language pathologist and physical therapist worked to support motor and social skills opportunities together, the likelihood that the child could carryover newly learned skills into real-life settings could improve. Although co-treatment between speech–language pathologists and physical therapists is often considered complementary and mutually supportive across both disciplines ([61]), co-treatment in practice is often limited ([57]). In response to our first research question, this study offers further evidence that social communication and balance skills are related in autistic children, and therapies that consider both would be beneficial.

#### 4.1.2. Age and Balance Abilities

In addition to autism characteristics, the first cultural variable found to be a significant predictor of the MABC-2 Balance scores was age. This was initially unexpected because the MABC-2 is split into three age bands with differing test items designed to assess similar constructs across each age band to account for typical motor changes across development ([31]). In other words, age should already be factored into test item selection and administration. However, other researchers have reported that the sensitivity and specificity of the MABC-2 differs across each of the different age bands in other movement conditions, such as developmental coordination disorder ([66]). In a large-scale study that included European and African children, the sensitivity in identifying children with developmental coordination disorder was only moderate for the youngest two age bands, specifically for Age Band 1, which spans ages 3–6 years, and Age Band 2, which spans ages 7–10 years ([66]). The authors of this previous study point out that this is especially problematic, as this is the age when children learn culturally influenced motor skills ([23]). Although the current study included autistic and non-autistic children (rather than children with developmental coordination disorder) within the United States (rather than in Europe or Africa), age was also found to be a contributing factor on balance-based task performance of the MABC-2. This finding further exemplifies how basic cultural factors, such as age, must be considered when diagnosing, assessing, and tailoring interventions to serve all children.

#### 4.1.3. Gender/Sex and Balance Abilities

Gender/sex was one cultural variable identified in this study that significantly predicted the MABC-2 Balance score. In this study, the pooled non-male (i.e., female, non-binary, and non-reported gender/sex) participants were more likely to obtain higher MABC-2 Balance scores than the males. This result is consistent with studies from Australia ([21]), China ([40]), Singapore ([62]), and South Africa ([59]), in which girls outperformed boys on the MABC-2. [59] ([59]) suggest that tasks such as threading a bead on a hook or jumping in a square, like playing hopscotch, are tasks that girls are inherently more skilled at or may be more likely to participate in based on cultural expectations. These sex differences have been tested in different countries using the MABC-2, revealing that in both the Manual Dexterity and, of most importance to this study, the Balance subtest, girls outperformed boys ([21]; [40]; [59]; [62]; [64]). Regardless, the MABC-2 does not have separate norms for females and males, which is a concern considering the apparent gender/sex-bias of the test. Because the same normative data is applied to children across genders or sexes, but males tend to have lower raw scores on the MABC-2, there is a greater chance of males being identified as showing motor difficulties using this test, whereas non-male children are more likely to go undetected as having any motor difficulties ([59]). Although the previously discussed results apply to non-autistic children, similar results in this study may indicate that gender/sex should be considered when assessing motor skills in autistic children as well.

In this study, the number of male and female participants were equal in the non-autistic group; however, the autistic group was heavily skewed male, with only one autistic female participant. Consequently, our finding that non-male participants showed higher balance scores (β = 2.92, *p* = 0.007) is driven primarily by non-autistic children and cannot be generalized to autistic females. According to the CDC, autism is 3.4 times more commonly diagnosed in males compared to females ([58]), which helps explain the gender/sex disparity of the autistic participants in this study. Females have historically been excluded from participation in autism research when inclusionary criteria utilized the ADOS-2 for diagnostic purposes due to the gender/sex biases it has shown ([37]). This study did not use the autism status on the ADOS-2 as an inclusionary criterion; however, it is likely that there are fewer autistic females with a diagnosis than males because of previous diagnostic use of the ADOS-2 or other gender/sex-biased evaluation tools. To address this gender/sex disparity, future research should engage in more targeted recruitment practices, such as sharing flyers with more female-dominant settings (e.g., with Girl Scout troops, dance or gymnastics centers, etc.), to reach more autistic females, ensure adequate representation, and allow for valid conclusions regarding gender/sex effects.

It is also worth noting that parent perceptions of their child’s autistic characteristics have previously been reported to correlate with the child’s balance scores ([27]), and if girls perform better than boys on standardized motor assessments ([59]), it is possible that their strengths in motor skill mitigate their parents’ concerns that their child may be autistic, thus delaying the parents’ pursuit of seeking an autism diagnosis for their daughter. Previous research has shown that autistic girls must present with more pronounced autism characteristics than their male counterparts to secure an official diagnosis ([32]), and that autistic females are more likely to engage in more masking, or camouflaging, to hide their autistic traits ([34]). Perhaps autistic females’ more advanced motor skills than their male peers contribute to their underdiagnosis compared to autistic males. However, because only one autistic participant in the current study identified as female, this study is unable to explore the relationship of autistic traits and balance skills in the female participants exclusively and therefore cannot shed much light on this possibility further.

#### 4.1.4. Ethnic, Social, and Cultural Effects

Finally, ethnic origin was revealed to be a significant predictor of a child’s MABC-2 Balance scores, in that the presence of reported Hispanic and/or white heritage was associated with higher motor performance. Importantly, the “Other” reported heritage groups (Black, Asian/Pacific Islander, American Indian/Alaskan Native) were pooled together for analysis, which may mask the heterogeneity of this sample. Furthermore, the heritage indicators were not mutually exclusive, so a single participant could have multiple ethnic heritage indicators included in the regression model at the same time. If a child reported multiple ethnic origins (e.g., white and Hispanic), then their predicted MABC-2 balance score would reflect the effects of both heritages added together. As such, the following possible interpretations are made cautiously, and these findings should be validated in larger, more diverse samples.

One possible explanation for the difference in motor performance based on the ethnic backgrounds of the participants could be due to the cultural alignment or mismatch between our study’s participants and those of the normative sample of the MABC-2, who resided in the United Kingdom, Canada, and the United States ([31]). For example, in a previous study comparing the validity of using the MABC-2 in a nationally representative sample of 2185 Chinese children, differences in motor performance emerged ([40]). Specifically, the children in the normative sample performed better in the MABC-2 Aiming & Catching tasks than their Chinese counterparts, whereas the Chinese children performed better in the Manual Dexterity and Balance tasks compared to their U.K. (normative sample) peers ([40]). As discussed previously, motor abilities and motor development are not the same for all cultures ([38], [39]), therefore norms that are based on one culture are not necessarily a good representation of what is considered “within [the] typical range” for children from other cultures.

Another possible explanation for the relationship between ethnic origin and motor abilities may be systemic, regarding the ethnic disparities in autism diagnosis. Historically, minority children have been under-identified for autism ([20]; [58]). Disparities in autism diagnosis for minority children may be explained by difficulties accessing healthcare and that minority children may have to show more pronounced autism characteristics in order to receive a diagnosis ([8]; [58]). Because motor abilities and autism characteristics are strongly related, we may assume that the more pronounced autism characteristics one has, the more motor difficulties they may have as well. The non-white participants in this study may have had to show more prominent autism characteristics to receive a diagnosis and therefore also had more pronounced motor difficulties than their white peers who could have received a diagnosis with fewer pronounced characteristics of autism.

It is worth reiterating that the overwhelming majority of participants in this study were white, despite various efforts to create an inclusive study with a variety of ethnic backgrounds, and that these results should be interpreted with caution. The research team includes individuals from various ethnic backgrounds; however, the geographic area in which this study was conducted is primarily white, which may explain (but does not excuse) this ethnic origin disparity in the sample. This study was conducted in DeKalb, Illinois, a rural area of the United States. The local demographics are roughly 62% white and white people make up the majority of the population that would have been recruited to this study ([63]). To try to overcome this ethnic disparity, recruitment materials were available in English and Spanish and were distributed at various community events and locations that cater to diverse populations in the area (e.g., libraries, daycare centers). To better represent the possible cultural differences with MABC-2 and autistic characteristics, more participants with diverse backgrounds should be recruited in future research by targeting areas of greater diversity such as nearby cities.

### 4.2. Moderation by Cultural Factors

Our second research question asked whether cultural factors moderate the relationship between autistic characteristics and balance abilities. After controlling for family-wise error rate using Holm–Bonferroni correction, we found no evidence that the tested cultural factors (gender/sex, ethnic origin, socioeconomic status, or multilingual exposure) significantly moderated the autism–balance association. This suggests that, within the range of the cultural variations represented in our sample, the negative relationship between autistic characteristics and balance performance was relatively consistent across groups.

The absence of significant moderation effects contrasts with theoretical expectations that cultural practices might differentially shape motor skill expression ([38], [39]). Several interpretations warrant consideration. First, our modest sample size (*N* = 44) limited power to detect small-to-moderate moderation effects (*f*^2^ < 0.15), even with appropriate statistical corrections. Sensitivity analyses indicated we were powered to detect moderate-to-large interactions (*f*^2^ ≥ 0.24) but likely missed smaller effects. While small moderation effects are less concerning from a practical standpoint, they could nonetheless be theoretically meaningful and merit investigation with larger samples.

Second, the intercorrelations among cultural variables (discussed below under Limitations) may have obscured independent moderating influences. When ethnic origin, SES, and number of languages are intertwined as they naturally are in families and communities, isolating the unique moderating effect of any single factor becomes statistically and conceptually challenging.

Third, balance abilities, particularly as assessed by the MABC-2 standardized tasks, may be less culturally malleable than other motor skills. The MABC-2 Balance subtest involves highly structured, constraint-driven activities (standing on one leg with arms at sides for 30 s, walking heel-to-toe in a straight line, jumping and landing in balance). These tasks differ from more culturally variable motor activities like free play, traditional games, or culturally specific movement patterns. Karasik and colleagues ([38], [39]) demonstrated cultural differences primarily in infant locomotor milestones (sitting, walking onset) where caregiver practices vary substantially. In contrast, once basic postural control develops, performance on constrained balance tasks may be less sensitive to cultural variation in motor experience.

Fourth, if the autism–balance association primarily reflects shared neurobiological substrates, specifically, cerebellar development as posited by the Cerebellar Sensitive Period Hypothesis ([69]), then cultural factors might influence absolute performance levels without substantially altering the strength of association between autistic traits and balance. This interpretation is consistent with our finding that cultural variables (age, gender/sex, ethnic origin) showed significant main effects on balance (Section 3.1) but did not moderate the autism–balance relationship. In other words, cultural factors may shift “where a child starts” on balance abilities without changing “how autism affects balance development.”

Finally, though no interactions reached significance in our sample, exploratory examination of effect sizes in the omnibus model (Section A.3) revealed the largest coefficients for SRS-2 × Hispanic Heritage (*β* = −3.37, *p* = 0.183) and SRS-2 × SES (*β* = −1.01, *p* = 0.117). Both negative coefficients could tentatively suggest that the autism–balance association might be slightly weaker in children from Hispanic heritage backgrounds or higher SES families. If replicable, such patterns could indicate that certain sociocultural contexts partially buffer against balance difficulties in autistic children. For example, children from higher SES (based on maternal education) have more access to early interventions than children from lower SES ([25]; [56]), which may mitigate the impact of the autism–motor relationship for higher SES children. However, given the limited sample and absence of corrected significance, these effect size patterns should not be interpreted as evidence, but rather as hypotheses for future testing.

### 4.3. Limitations

#### 4.3.1. Sample Size and Power Constraints

Although our sample (*N* = 44) exceeded the a priori requirements for detecting moderate main effects, the sample size limited our statistical power to detect small-to-moderate interaction effects (*f*^2^ < 0.15) in the moderation analyses. Thus, the absence of significant interactions should be interpreted as an absence of detectable moderation under current power, rather than as evidence that cultural moderation does not exist. Despite these constraints, this study establishes the feasibility of culturally informed autism–motor research and confirms that autistic characteristics and balance remain strongly associated when cultural factors are covariates. These findings serve as pilot evidence to motivate future large-scale investigations with culturally balanced cohorts, which are necessary to detect subtle moderation patterns and further disentangle the complex interplay of culture and motor development.

#### 4.3.2. Overlap Among Cultural Variables and Sample Characteristics

Correlation analyses revealed substantive intercorrelations where Hispanic and “Other” ethnic origins were associated with higher SRS-2 scores (*r* = 0.30 for Hispanic and *r* = 0.32 for Other), and higher SES was associated with lower SRS-2 scores (*r* = −0.30). This pattern likely reflects documented diagnostic disparities, wherein ethnic minority and economically disadvantaged children often require more pronounced autistic traits to receive a diagnosis ([20]; [58]). Consequently, our sample may over-represent more prominent autistic features among minority participants, potentially confounding cultural main effects with sampling artifacts. Although variance inflation factors indicated acceptable multicollinearity (VIF < 1.6), the interdependence of ethnic origin, SES, and language restricts our ability to isolate the unique contribution of any single cultural factor. Future research should include larger, more diverse samples to disentangle the contributions of these cultural dimensions, but this study lays the groundwork justifying the need for this future research.

#### 4.3.3. Research Team Composition and Community Representation

Finally, it is worth noting that no members of this research team identify as autistic. The inclusion of an autistic researcher on this team would have been beneficial to the project in that it would have followed a community-based research approach. Community-based research includes research team members that are of the same community as the participants ([68]). The main benefits of community-based research are increase in participant trust and decrease in “business as usual” research practices dominated by white, educated, financially powerful individuals ([68]). Because no autistic researchers were on this team, the potential benefits offered by having an autistic researcher’s unique perspective are therefore limited in this study. Future research should include autistic researchers on the team to represent the community participating in the study.

## 5. Conclusions

Autistic traits and balance skills are highly related to one another. This study confirms that this relationship remains robust even when controlling for cultural variables. Culture influences perceptions of autism ([16]; [18]; [30]; [35]) and motor skill development ([38], [39]), yet the interaction between autism, culture and motor skills taken together is rarely considered in autism research and diagnostic tools ([11]; [55]; [70]). Our results demonstrate that cultural factors, such as gender/sex and ethnic origin, significantly predict balance outcomes on a commonly used childhood motor assessment. To increase the accuracy and representation of diagnostic tools and future treatment targets, cultural factors must be considered in the development and implementation of diagnostic and treatment protocols. Additionally, it is necessary for autism diagnostics and treatment to include physical therapy professionals to improve quality and effectiveness of treatment, as well as considering the whole child. Autism research that includes children from more diverse backgrounds is necessary to continue to develop and use culturally responsive practices that represent the individuals being served.

## Figures and Tables

**Figure 1 behavsci-15-01742-f001:**
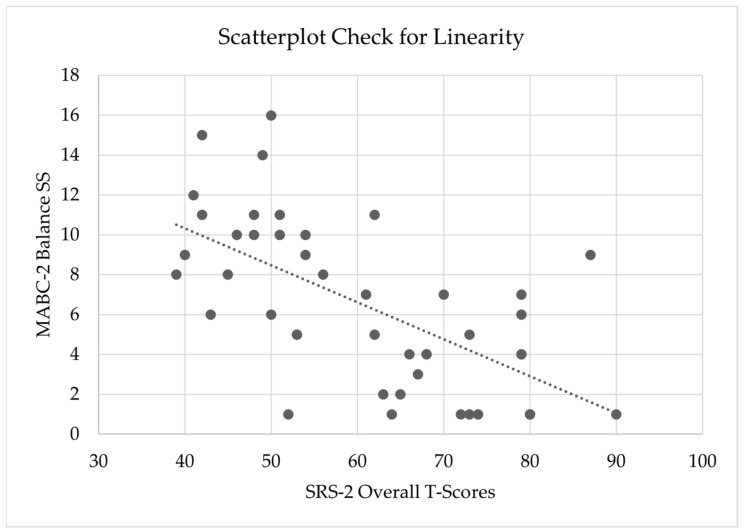
Scatterplot of MABC-2 Balance Scores and SRS-2 *T*-scores to check for linearity.

**Table 1 behavsci-15-01742-t001:** Descriptive Measures of Participants and Assessment Scores.

**Measure**	** *n* **		
Gender/Sex *			
Male	27		
Female	15		
Nonbinary	1		
Not Reported	1		
Number of languages			
1	39		
2	5		
Ethnic Origin *			
American Indian or Alaskan Native	0		
Asian or Pacific Islander	0		
Black	3		
Hispanic	6		
White	28		
Hispanic & white	3		
Asian or Pacific Islander & white	3		
American Indian or Alaskan Native & Hispanic & white	1		
**Measure**	**Mean (SD)**	**Minimum**	**Maximum**
Age (years)	8.23 (2.25)	5	12
SES (Caregiver education in years)	17.45 (6.41)	0	33
MABC-2 Balance Subscale Score	6.50 (4.15)	1	16
SRS-2 *T*-score	60.64 (14.05)	39	90

* Due to small cell sizes, gender/sex was grouped as male (reference) vs. non-male (female, non-binary, or not reported). Ethnic origin was encoded with presence/absence indicators for white, Hispanic, and other heritage (pooled), reflecting that multiple heritages may be reported by the same participant.

**Table 2 behavsci-15-01742-t002:** Regression Coefficients for Variables Contributing to Movement Abilities.

Variable	*β*	95% CI	*t*	*p*
SRS-2 Overall *T*-score (*z*)	−3.00	[−3.95, −2.05]	−6.40	<0.001
Age (yrs; *z*)	1.61	[0.70, 2.53]	3.56	0.001
Gender_nonMale	2.92	[0.87, 4.98]	2.89	0.007
Ethnic origin_white heritage	5.61	[2.50, 8.72]	3.66	<0.001
Ethnic origin_Hispanic heritage	4.87	[2.22, 7.53]	3.73	<0.001
Ethnic origin_Other heritage	2.05	[−0.56, 4.66]	1.59	0.12
SES (C1 education yrs; *z*)	0.04	[−1.02, 1.10]	0.08	0.94
Number languages (*z*)	−0.21	[−1.09, 0.66]	−0.49	0.62

Note. Model fit: *R*^2^ = 0.6704; adjusted *R*^2^ = 0.5951; RMSE = 2.356; MAE = 1.783; AIC = 218.28; BIC = 234.34; Durbin–Watson = 2.257; Jarque–Bera *p* = 0.982. *p*-values shown as <0.001 when smaller than 0.001. Std. Error computed from 95% CI width as (Upper − Lower)/(2 × 1.96).

**Table 3 behavsci-15-01742-t003:** Moderation Analysis: One-by-One Interaction Tests.

Moderator	*β*	95% CI	*p*	*p_adj*	Δ*R*^2^	Model *R*^2^
Gender/Sex (nonMale vs. Male)	−1.11	[−3.35, 1.14]	0.323	1.000	0.009	0.680
White Heritage	0.16	[−2.16, 2.47]	0.890	1.000	0.000	0.671
Hispanic Heritage	−0.70	[−2.89, 1.49]	0.522	1.000	0.004	0.674
Other Heritage	−0.81	[−3.78, 2.16]	0.583	1.000	0.003	0.673
SES (Caregiver Education)	−0.46	[−1.50, 0.58]	0.376	1.000	0.008	0.678
Number of Languages	−0.55	[−1.46, 0.35]	0.223	1.000	0.014	0.685

Note. *N* = 44. *β* = unstandardized regression coefficient for SRS-2 Overall *T*-score (standardized) × Moderator interaction term; CI = 95% confidence interval; *p* = uncorrected *p*-value; *p_adj* = Holm–Bonferroni adjusted *p*-value controlling family-wise error rate at *α* = 0.05 across 6 tests; *ΔR*^2^ = change in *R*^2^ relative to baseline model (*R*^2^ = 0.670, including main effects of SRS-2, Age, Gender/Sex, ethnic origin heritage indicators, SES, and number of languages); Model *R*^2^ = *R*^2^ for model including the specific interaction. All continuous variables (SRS-2, Age, SES, Number of Languages) were *z*-scored prior to creating interaction terms. After adjustment for multiple comparisons, no interactions reached statistical significance. Taken together, none of the cultural variables tested moderated the autism–balance association, supporting the hypothesis that this relationship is relatively stable across the cultural dimensions represented in this sample.

## Data Availability

The datasets presented in this article are not readily available because the data are part of an ongoing study. Requests to access the datasets should be directed to Allison Gladfelter at agladfelter@niu.edu (the corresponding author).

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
