# Peer review of "Relationships Between Cultural Factors and Motor Abilities in Autistic and Non-Autistic Children"

_behavsci, 2025, doi:10.3390/bs15121742_

Round 1
Reviewer 1 Report
Comments and Suggestions for Authors
This paper investigates links between autistic traits balance performance and several “cultural” variables in a Midwest USA sample of children. Results show higher autistic traits are associated with weaker balance, age improves scores, girls and some heritage categories show better outcomes. Authors argues that cultural context matters for assessment and intervention.
I like this study. The question is relevant and practical for clinicians and special educators. Analysis is in general adequate and transparent. The writing is clear enought to follow even for practitioner. I especially appreciate the cautious tone about. For publication I reccomend minor revision with some focused improvements.
You formulate Research Question 2 about moderation by cultural factors. However, the regression includes main effects only. Please add interaction terms. If power is a concern, state a priori limit and report it. You can include interactions one by one with adjusted alpha, or provide an exploratory table in Appendix. Even a brief robustness check would make RQ2 answered properly.
The total N is small and there is only one autistic girl. This makes sex/gender effects very fragile. I absolutely understand the real-life difficulty of recruitment, especially with autistic children (I work in this field and know it is hard and slow). Still, please reflect this limitation more clearly in Discussion and when interpreting gender findings (avoid too strong claims).
Because you use only the Balance subtest, note that conclusions are about balance specifically, not “motor skills” globally. One sentence more explicit would prevent over-reach. If you can, add a sentence why Balance is theoretically most relevant for autistic traits in your framework.
If you add a brief moderation analysis for RQ2 (even limited/supplementary), strengthen the Discussion around sample limitations (especially the single autistic girl), and clarify coding/interpretation of heritage, the manuscript will be a useful and timely contribution. The core signal (higher autistic traits - poorer balance) is consistent with clinical experience and prior literature, and your cultural lens is valuable for equitable assessment.
Reviewer 2 Report
Comments and Suggestions for Authors
Peer Review, for Behavioral Sciences:Relationships between Cultural Factors and Motor Abilities in Autistic and Non-Autistic Children;
General Comments
This study on the motor skills of children with autism is based on select data taken from a larger study. It needs work in several ways. First, please refer to children with autism in the title and throughout instead of autistic children. Please refer to the relationship of the findings to the concept of ASD per the DSM-5-TR, Age and gender are not cultural factors, they are demographics; correct throughout. The sample is too small for the purposes despite the power analyses, as the overlap in the cultural variables suggests; something seems amiss there; perhaps after correction, you should refer to this as a pilot study. Describe the original study from which your data is drawn and show how the details of your study differs from the original study. Define clearly your hypotheses, or state the study is exploratory.
Specific Comments
Se the yellow highlights for specific changes. I do not refer to some in that I was uncertain if the indicated yellow highlight needed a comment, Some yellow highlights refer to the general comments above. Some are not indicated by line below because I given generic comments first. Refer to the beta symbol, not B. There are too few key words. Define abbreviations on first use, and use them throughout as their mention proceeds. Use “in that” instead of “as.” Use “that” instead of “which” if there is no comma after the which. Use “that” instead of “who” after the word “professions”. Use “it is” instead of it’s. In psychology, we italicize stat terms like N and z.
Line 118-9. Use not only limited to the effects of the … but also … criteria, as
123 Use rocking and, overall, [I think, not sure]
128 conducted a review – specify of what
140 disregard the yellow
158 Use countries,
170 Research - specify of what
177 Use “advantage” not benefit
185 but there are no goals related to social!
209 Delete “and”
Pages 8-9 Are the highlighted stats significant? Were they accommodated in the subsequent stats
- Psychology would use normative, nit norming, here and throughout
406-8 Put the comma after the parens
825 Use White and Whites make

Round 2
Reviewer 2 Report
Comments and Suggestions for Authors
Dear editors and authors
All requested changes have been addressed. Thanks for the attention to my comments and those of the other reviewer